# Examining the effects of transcranial direct current stimulation on human episodic memory with machine learning

**Aleksandra Petrovskaya**[1,2☯¤]**, Bogdan Kirillov**[3☯]***, Anastasiya Asmolova**[1,2]**, Giulia Galli**[4]**, Matteo Feurra**[1,2‡]**, Angela Medvedeva**[5‡]

**1** Psychology Department, National Research University Higher School of Economics, Moscow, Russian Federation, **2** Institute of Cognitive Neuroscience, National Research University Higher School of Economics, Moscow, Russian Federation, **3** Center of Life Sciences, Skolkovo Institute of Science and Technology, Skolkovo, Russia, **4** Department of Psychology, Kingston University, Kingston Upon Thames, United Kingdom, **5** Vivian L. Smith Department of Neurosurgery, University of Texas Medical School at Houston, Houston, Texas, United States of America

☯ These authors contributed equally to this work.
¤ Current address: Engelhardt Institute of Molecular Biology, Russian Academy of Sciences, Moscow, Russian Federation
‡ These authors share senior authorship on this work.
* Bogdan.Kirillov@skoltech.ru

**Data Availability Statement:** All related files are available from the Zenodo database (DOI: 10.5281/zenodo.3837111).

## Abstract

We aimed to replicate a published effect of transcranial direct-current stimulation (tDCS)-induced recognition enhancement over the human ventrolateral prefrontal cortex (VLPFC) and analyse the data with machine learning. We investigated effects over an adjacent region, the dorsolateral prefrontal cortex (DLPFC). In total, we analyzed data from 97 participants after exclusions. We found weak or absent effects over the VLPFC and DLPFC. We conducted machine learning studies to examine the effects of semantic and phonetic features on memorization, which revealed no effect of VLPFC tDCS on the original dataset or the current data. The highest contributing factor to memory performance was individual differences in memory not explained by word features, tDCS group, or sample size, while semantic, phonetic, and orthographic word characteristics did not contribute significantly. To our knowledge, this is the first tDCS study to investigate cognitive effects with machine learning, and future studies may benefit from studying physiological as well as cognitive effects with data-driven approaches and computational models.

## 1 Introduction

Transcranial direct-current stimulation (tDCS), a method of safely and non-invasively delivering a weak electric current through the cortex, has been gaining increasing attention [1] as a tool for studying and possibly enhancing episodic memory [2]. Anodal tDCS (atDCS) involves increasing cortical excitability in a target region such as the dorsolateral prefrontal cortex (DLPFC), which may in turn facilitate or enhance memory performance by decreasing

**Funding:** The article was prepared within the framework of the HSE University Basic Research Program and funded by the Russian Academic Excellence Project '5-100'. The funders had no role in study design, data collection and analysis, decision to publish, or preparation of the manuscript.

**Competing interests:** The authors have declared that no competing interests exist.

reaction time and/or increasing memory accuracy [3, 4]. However, there remains well-founded skepticism about widespread applications of tDCS, partly because of the lack of knowledge about mechanisms of action [5] and difficulty in replicating results [6].

Most tDCS studies in episodic memory have targeted the DLPFC (see [7] for a review) because of its role in selective attention [8], strategic retrieval [9], and other executive functions involved in episodic memory [10]. In line with the hemispheric encoding/retrieval asymmetry model [11], previous studies with tDCS show functional asymmetry in the DLPFC, with stimulation over the left DLPFC affecting encoding and stimulation over the right DLPFC affecting retrieval [12–14]. However, memory enhancements due to atDCS have been inconsistent over the DLPFC, in both individual studies [4, 10, 12, 13, 15] and meta-analyses [16–18]. Most studies over the DLPFC examined effects when atDCS was delivered predominantly during encoding. At least two studies examined purely offline encoding effects: [19] delivered atDCS over F3 for 15 minutes before encoding and found no effects on accuracy or reaction time, although atDCS over the contralateral hemisphere (F4) led to faster reaction times. Lu and colleagues [20] delivered atDCS over a more lateral site (FC5) for 20 minutes before encoding and found an increase in correct memory responses for previously-presented items.

To the authors' knowledge, no study to date has systematically examined effects of timing of administration on the DLPFC, comparing online vs offline effects at encoding or retrieval. However, a previous study by the authors showed significant differences in online and offline effects over an adjacent site, the left VLPFC [20]. The authors administered atDCS before the study phase (offline encoding) or during the study task (online encoding) that involved intentionally memorizing each presented word (Experiment 1). As expected given support from fMRI and TMS studies, the left VLPFC seemed to be strongly modulated by atDCS at encoding, specifically online but not offline encoding.

The VLPFC and DLPFC are thought to play functionally distinct roles in long-term verbal memory, and the left VLPFC appears selectively engaged in verbal but not non-verbal material. Specifically, the VLPFC may be more involved in encoding for individual items, while the DLPFC is more engaged in associative or relational encoding [21–24]. Moreover, activation in the DLPFC may predict long-term memory success through DLPFC involvement in domain-general working-memory or executive processes such as mental manipulation of information (e.g. visualizing rotating objects; [25]) and applying a specific strategy (e.g. for retrieval of information; [26]). In a tDCS study conducted by [27], tDCS was applied over the left DLPFC in a cued-word-completion task to modulate memory performance. The results showed that cathodal stimulation inhibited memory accuracy after error learning in comparison with sham: the errors were evoked through guessing, or the incorrect completion lowered memory accuracy in contrast to a non-error condition or anodal stimulation in either condition (no effect). They concluded that the DLPFC was only modulated when its processing demands were higher for conflicting information. In contrast, the VLPFC appears to be engaged in maintaining information in working memory [28] and processing semantic information including accessing lexical identity and connecting words to other words in the semantic network [29].

The current study involved administering atDCS over the left DLPFC during offline or online encoding to identify the effective time of administration and confirm the causal role of the DLPFC in encoding processes. Participants were assigned to a Sham group (over the DLPFC) or one of three tDCS groups: DLPFC Online, DLPFC Offline, and VLPFC Online. Previous research suggests that atDCS could potentially enhance memory performance when delivered online or offline encoding over the DLPFC [30–34], so we predicted higher memory performance for either condition (DLPFC Online or DLPFC Offline). In addition, the study provides a novel comparison of atDCS effects during encoding over the VLPFC vs DLPFC.

We predicted a successful replication of Experiment 1 from [20] with higher memory accuracy for the VLPFC Online group compared to Sham, but we also predicted a larger effect size for VLPFC Online than DLPFC Online or DLPFC Offline based on previous rTMS studies comparing memory disruption in the VLPFC vs DLPFC. For example, [35] found that stimulation over the VLPFC during encoding led to a greater disruption in memory performance than over the DLPFC, suggesting that the VLPFC may play a more important role in encoding processes. Finally, there remains a lack of knowledge about the mechanisms of atDCS on verbal memory and whether atDCS effects can be influenced by word characteristics, specifically semantic (meaning), orthographic (letters), and phonological (sounds). We also aimed to examine how the tDCS effects interacted with language and linguistic factors, since the replication study was conducted in Russian and the original study was conducted in English. Thus, data were applied from a previously-published experiment ([20]; Experiment 1) and the current experiment from the Online Encoding (VLPFC) group to an AUROC analysis. After comparisons of group means in reaction time and accuracy on the recognition test, a regression algorithm was applied to infer the quality of recognition using semantic and phonological features of words. Semantic and phonological features were added to the model and predictive capability was assessed. We predicted that the model would be able to distinguish between words that are hard and easy to remember in Russian and English speakers. Thus, machine learning algorithms were applied in order to reveal the impact of factors including semantics, phonetics and individual variance to episodic memory performance as well as the interaction with tDCS.

## 2 Methods and materials

In line with a replication study, we followed the same procedure as in Experiment 1 of [20] with few exceptions (translation of materials to Russian and comparison of DLPFC and VLPFC as stimulation sites). Broadly, participants memorized words presented individually on a screen while undergoing tDCS (before or during this phase), and 24 hours later they performed a recognition test. The study and test stimuli were translated based on the first word meaning in the vocabulary entry, and the translation achieved relatively matched frequency ($M_{English}$ = 24.47; $M_{Russian}$ = 40.38; [36]) and number of letters ($M_{English}$ = 6.17; $M_{Russian}$ = 6.19). Specifically, we applied the same tDCS settings (2 mA active tDCS and 30 s Sham tDCS) over the DLPFC (all conditions: offline, sham, and online) in addition to the VLPFC (online encoding only). The electrodes were placed at both sessions (study and test 24 hours later), but stimulation was only delivered before or during the encoding task in the study phase. The anode was placed over the left VLPFC (F7) or the left DLPFC (F3), and the cathode was placed over the ipsilateral (left) shoulder. See Section 2.1 below for the experimental conditions. Stimulation was delivered via a battery-powered, constant-current stimulator (Brainstim, EMS, Bologna, Italy) through $5 \times 7 cm^2$ electrodes. In addition, participants were asked to describe the sensations that they felt during the stimulation and indicate whether they believed they received real or placebo stimulation. Participants were single-blinded, and the questionnaire indicated that blinding was successful: 80% believed that they received real stimulation. Data were analyzed in SPSS (version 24; IBM, Armonk, New York) and machine learning studies were conducted in Python (version 3.7).

### 2.1 Participants

Participants were randomly assigned to one out of four groups: VLPFC Online, DLPFC Online, DLPFC Offline and Sham. Based on a power analysis to detect a large effect size [20], $d = 1.29; \alpha = 0.05, 1 - \beta = .95$), we aimed to recruit 31 participants per group (124 total), but we

analyzed data from 97 participants because of exclusions (see below). Participants (female = 63, male = 49, $M_{age}$ = 20.51, SD = 2.89) were native Russian speakers with normal or corrected-to-normal vision and no history of neurological or psychiatric illness. Bonferroni-corrected pair-wise comparisons of ae between pairs of groups showed a significant difference in age between VLPFC Online (M = 21.76, SD = 3.54) and DLPFC Online (M = 19.21, SD = 1.10), p = .014. However, there were no significant differences in age between the other groups, $p_s$ > .062.

Data from 12 participants were excluded from analysis because 1) the participant was feeling unwell (2 participants: DLPFC Sham and VLPFC Online) 2) there were technical issues (6 participants: two in DLPFC Online, two in VLPFC Online, and two in DLPFC Sham) or 3) the participant was left-handed (one participant: DLPFC Online) and 4) there was an experimenter error (3 participants: one in DLPFC Sham and two in VLPFC Online). The exclusions resulted in 26 in the VLPFC Online group, 31 in the DLPFC Offline group, 28 in the DLPFC Online group, and 27 in the Sham group. Moreover, fifteen outliers were excluded (see data analysis for criteria), leaving a final sample size of 25 for VLPFC Online (one outlier), 25 for DLPFC Offline (six outliers), 23 for DLPFC Online (five outliers), and 23 for DLPFC Sham (four outliers). For all Machine Learning analysis all DLPFC Offline participants except one were used (30 participants).

The study was approved by the ethics committee of National Research University Higher School of Economics (Moscow, Russia) and followed the corresponding ethical guidelines. All participants provided written informed consent and were given monetary compensation (500 rubles) for their time.

## 2.2 Procedure and experimental design

Each participant, regardless of group assignment, came to the laboratory twice within a 24 hour-interval: on the first day participants memorized the verbal stimuli one word at a time through pleasantness judgements (pleasant or unpleasant), and on the second day they performed an old/new recognition memory task. For DLPFC Sham, stimulation was delivered for 30 seconds before the stimulator was turned off, and the start of the stimulation corresponded to the start of the reading task for half the Sham participants and the start of the memory task (study phase) for the other half. Stimulation was applied for the entire duration (10 minutes) of the reading task for the DLPFC offline group and for the entire duration (10 minutes) of the study memory task that immediately followed for the VLPFC Online and DLPFC Online groups. In summary, the stimulation was delivered for approximately 10 minutes to cover the entire duration of the reading and memory tasks.

## 2.3 Data analysis

One-way ANOVAs were conducted for each measure of recognition accuracy [37]: discrimination index (Pr: combined index involving hits and false alarms), proportion of hits (correctly identifying an "old" item as studied), and proportion of false alarms (incorrectly identifying a new item as studied). In addition, one-way ANOVAs were conducted for average reaction times and reaction times for hits and false alarms separately. Finally, a one-way ANOVA was conducted on response bias, an index involving hits and false alarms (higher values indicate a more conservative pattern of responding to old items). Significant effects were followed up with Bonferroni-corrected planned contrasts between each stimulation group and Sham (one-tailed). Outliers were excluded based on two standard deviations from the mean on any of the following dependent measures: discrimination index Pr, proportion of hits, proportion of false alarms, and reaction times for hits or false alarms.

## 2.4 Area under the receiver operating characteristic curve as a measure of predictive performance for episodic memory changes

For the second analysis including data from the [20], we included all 17 participants from the Sham group and 17 from the Online tDCS group, resulting in the same mean age with no significant differences between the groups. From the current study, we only included data from the DLPFC Sham (N = 23) and VLPFC Online groups (N = 25) with no significant differences in age between groups (Bonferroni-corrected paired comparisons), $p_s > .852$. We implemented a 2 × 2 design with the first factor as language (English or Russian) and the second factor as tDCS group (VLPFC or Sham).

Applying Area Under the Receiver Operating Characteristic Curve (AUROC) was appropriate given that the task involved binary classification (coded as 1 if the word was presented at study and 0 if unpresented), which fits any ML approach to measuring predictive performance of classification models [38]. In ML Classification models, true positives represent the model predicting the positive class correctly and false positives represent incorrect prediction of positive class.

AUROC 1) provides information about true and false positives in a single measure 2) shows the source of model error, with larger values indicating that the model predicts better than chance (greater true positives) and 0.5 indicating the opposite and 3) does not assume a normal distribution and is robust to unequal sample sizes (whereas average accuracy measures would overestimate performance). Thus, it is the most suitable measure for the current data.

We included a trial-based approach, computing AUROC for each word based on the entire sample's responses for that word, and a participant-based approach, computing AUROC for each participant based on responses (old or new) for all words presented to that participant.

AUROC values can show whether tDCS is effective in enhancing memory performance, and the subsequent ML analyses using AUROC can show the effects of various factors (semantic, phonetic, orthographic word characteristics) on recognition of individual words. We attributed any differences in memory performance not explained by differences in sample size, linguistic characteristics, or group (tDCS vs Sham) to individual differences in memory function.

If stimulation enhances episodic memory performance, we expect a significant increase or decrease in AUROC (both over words and over participants in both VLPFC groups compared to Sham). If tDCS impairs memory performance, we expect values close to 0.5, which reflects that participant response decisions (old or new) approached chance.

If tDCS has no effect, there would be no difference in AUROC values across words or AUROC distributions between VLPFC and Sham. Since tDCS enhanced episodic memory in the original experiment and we expect a successful replication, we expected AUROC values for VLPFC to be significantly different from chance, with significantly different AUROC values and distributions compared to Sham. Moreover, tDCS significantly increased reaction time in [20], so we predicted a significant difference between VLPFC and Sham in reaction time. Since we did not assume a normal distribution, we applied the two-sample Kolmogorov-Smirnov test for accuracy and reaction time. We have also computed median test for differences in median AUROC.

## 2.5 Extraction of semantic, phonological and orthographic determinants of episodic memory performance

To investigate the effect of semantic and phonetic features on memorization and the interaction with tDCS, we extracted words with significantly more true positive rates (reflecting hits) and false positive rates (reflecting false alarms) in each group (VLPFC and Sham). We

examined the top ten words from each category (see S1 Appendix). A preliminary descriptive analysis of the highest and lowest AUROC-words showed no association between individual variance and memory accuracy in VLPFC and Sham groups. Therefore, we do not report these results further.

Moreover, we used word embeddings, which approximated the semantic similarities between words. In line with the trial-based and participant-based approaches, we conducted two kinds of ML studies: 1) Participant-independent (trial-based) analysis tests the success of a model that predicts the AUROC for each word using either word embeddings or letter-based one-hot encoding (each letter $i$ is replaced by a vector of $N_{alphabet}$ components with the $i$-th component being equal to 1 while every other component is zero.) and 2) Participant-specific (participant-based) analysis tests the success of a model that predicts whether a participant was able to recall the word given either word embeddings or letter-based one-hot encoding for each word for individual participants. Word embeddings were applied from FastText [39], a library developed in Facebook that incorporates semantic information from each word as well as subwords contained within to embed vectors. The pretrained embedding was aligned following [40] so that English and Russian vectors could lie in the same vector space and we could use it to train a model that uses both English and Russian vectors to check whether there are any language-independent semantic determinants. To artificially enlarge the sample size for our models, we use the augmentation approach described in [41] (see S2 Appendix for details).

For prediction, we applied TPOT [42], an automated Machine Learning library that enables searching for the best classification and regression model using Evolutionary Algorithms, and AutoPyTorch [43], an automating Machine Learning library for the PyTorch [44] deep learning framework. Pipelines found by TPOT or AutoPyTorch are usually better than those found by grid search or manual construction. TPOT was used for experiments with word vectors, and AutoPyTorch was used for experiments with one-hot encoded word images.

We applied the following parameters: for TPOT, default ones for evolutionary algorithm; 20 for number of generations and 5 for population size; 10 folds for cross-validation; mean absolute error as the scoring function, and "TPOT light" as the configuration dictionary, for AutoPyTorch, default validation setup, "tiny-cs" config for participant-independent trials and "medium-cs" config for participant-dependent trials.

Prior to training, for each experiment, we have randomly extracted 10% of the dataset and used it as a hold-out set for test. We trained our AutoML on the rest 90% with the respective cross-validation schemes and we examine the generalization ability of a model by assessing the difference between the quality metrics on training and hold-out sets. The model could perform very well on training set, but show lack of predictive ability on the hold-out. Such an outcome would be a sign of poor generalization ability which would show that based on this data we can not infer the existence of learnable connection between features and a label.

## 3 Results

### 3.1 ANOVA

Accuracy was significantly above chance for all groups, $t_s > 3.88$, $p_s < .005$. Levene's test ($F_s > 2.84$, $p_s < .042$) and frequency distributions suggested that most data did not follow a normal distribution. Although non-parametric statistics may be more appropriate than an ANOVA, to enable comparison with the original effect sizes, corrected parametric statistics (Brown-Forsythe's F statistic) are reported in text for corresponding dependent measures. There were no significant differences between groups in recognition accuracy $F(3, 79.78) = 2.37$, $p = .076$, $\eta_p^2 = .082$, response bias $F(3, 93) = 2.15$, $p = .100$, $\eta_p^2 = .065$, or average reaction time

**Table 1. Means and standard deviations for memory accuracy across groups.**

|                       | N  | Discrimination Pr | Br response bias | Pr hits | Pr false alarms |
|-----------------------|----|-------------------|------------------|---------|-----------------|
| **Online DLPFC mean** | 24 | 0.15              | 0.61             | 0.68    | 0.53            |
| **Online DLPFC SD**   | 24 | 0.16              | 0.14             | 0.10    | 0.18            |
| **Offline DLPFC mean**| 25 | 0.07              | 0.60             | 0.63    | 0.57            |
| **Offline DLPFC SD**  | 25 | 0.12              | 0.12             | 0.12    | 0.14            |
| **VLPFC mean**        | 23 | 0.16              | 0.56             | 0.63    | 0.47            |
| **VLPFC SD**          | 23 | 0.18              | 0.17             | 0.16    | 0.18            |
| **Sham mean**         | 25 | 0.09              | 0.66             | 0.69    | 0.59            |
| **Sham SD**           | 25 | 0.11              | 0.12             | 0.11    | 0.12            |

$F(3, 74.26) = 0.70, p = .556, \eta_p^2 = .028$. To enable comparison with the original paper, we report the results of multiple ANOVAs. However, to control for multiple tests and the difference in age between groups, we analyzed all dependent variables including AUROC with a MANOVA, Pillai's trace $(12, 294) = .166, p = .148$, and recognition accuracy and average reaction time with age as a covariate, $F(3, 98) = .097$ and $F(3, 98) = .557$, respectively. There was no significant difference between groups with hits, false alarms, recognition accuracy, and average reaction time as dependent variables. There was also no significant difference in the proportion of hits $F(3, 76.88) = 1.37, p = .259, \eta_p^2 = .051$, or associated reaction times $F(3, 75.56) = 0.49, p = .694, \eta_p^2 = .019$. There was a significant difference in proportion of false alarms $F(3, 93) = 2.85, p = .042$ but not associated reaction times $F(3, 70.82) = 0.91, p = .439, \eta_p^2 = .037$. Planned contrasts for false alarms revealed significant differences betzween VLPFC tDCS and Sham $t(93) = 2.78$ and between Offline DLPFC tDCS and Sham $t(93) = 2.14, p = .018$, with higher false alarm rates for VLPFC tDCS and Offline DLPFC tDCS. There were no significant differences between Online tDCS and Sham $t(93) = 1.36, p = .178$. See Table 1 for means and standard deviation for the mean accuracy for each group and Table 2 for means and standard deviation for reaction times.

## 3.2 AUROC analysis and ML models

The range of performance as measured by AUROC (0.38-0.62; see Fig 1) indicates that participant decisions were close to random choice, since an AUC coefficient of 0.75 or higher reliably reflects accurate performance. The VLPFC group shows a similar distribution of individual AUROC coefficients to the Sham group, indicating that there was no effect of tDCS on memorization and perhaps other factors (word characteristics, individual differences, error) contributed more highly.

**Table 2. Means and standard deviations for reaction time accuracy across groups.**

|                       | N  | Average RT | RT hits | RT false alarms |
|-----------------------|----|------------|---------|-----------------|
| **Online DLPFC mean** | 24 | 501.49     | 504.82  | 498.17          |
| **Online DLPFC SD**   | 24 | 144.91     | 151.20  | 139.84          |
| **Offline DLPFC mean**| 25 | 503.00     | 505.85  | 500.16          |
| **Offline DLPFC SD**  | 25 | 103.62     | 104.71  | 103.66          |
| **VLPFC mean**        | 23 | 535.72     | 532.52  | 538.91          |
| **VLPFC SD**          | 23 | 156.16     | 142.72  | 171.43          |
| **Sham mean**         | 25 | 484.00     | 477.82  | 484.00          |
| **Sham SD**           | 25 | 82.58      | 86.10   | 82.58           |

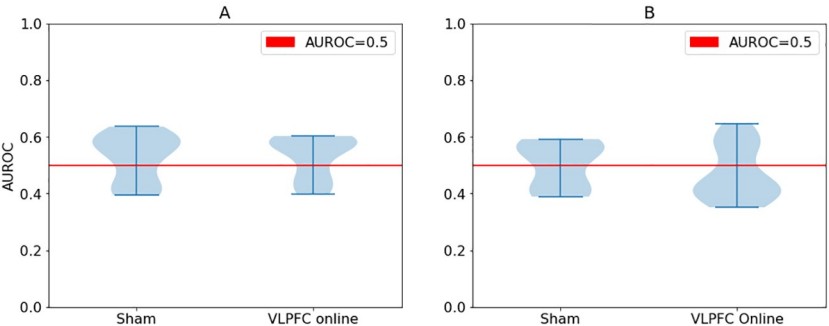

**Fig 1. Violin plots of AUC distributions for Russian vs English participants in each group.** Panel A shows the Russian sample (p-value of median test is 0.267) and Panel B shows the English sample (p-value of median test is 0.17).

Although the distributions of AUROC coefficients for individual words appeared to vary depending on the sample (English or Russian) and group (VLPFC or Sham; see Fig 2), with the English VLPFC group showing more variance than either of the Sham groups, there was no significant difference in AUROC distributions between English or Russian words or experimental groups, suggesting that the greater variance in the English group reflected smaller sample size.

We found significant differences in distributions for reaction time between Russian and English words as well as significant differences in distributions between sham and VLPFC online across both samples, $p < .001$, but no significant difference in median AUROC (p-values are either way larger than 0.001 or borderline—about 0.001-0.004), suggesting that as for the accuracy data, differences in distribution reflected differences in sample size (see Fig 3).

Finally, we found no significant contributions of semantic, orthographic, or phonological characteristics of words in terms of predicting the success of participant in recalling the word for participant-independent (see S3 Appendix) or participant-dependent (see S4 Appendix) models. The Spearman correlations for predictions and real labels for hold-out set in

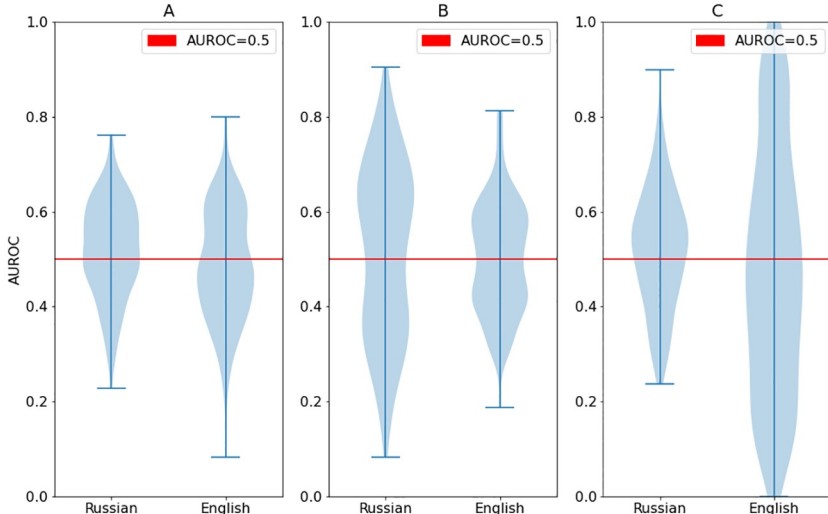

**Fig 2. Violin plots showing AUROC distributions for Russian vs English words in each group.** A—all groups (p-value of median test is 0.004), B—sham (p-value of median test is 0.274), C—VLPFC online (p-value of median test is 0.001).

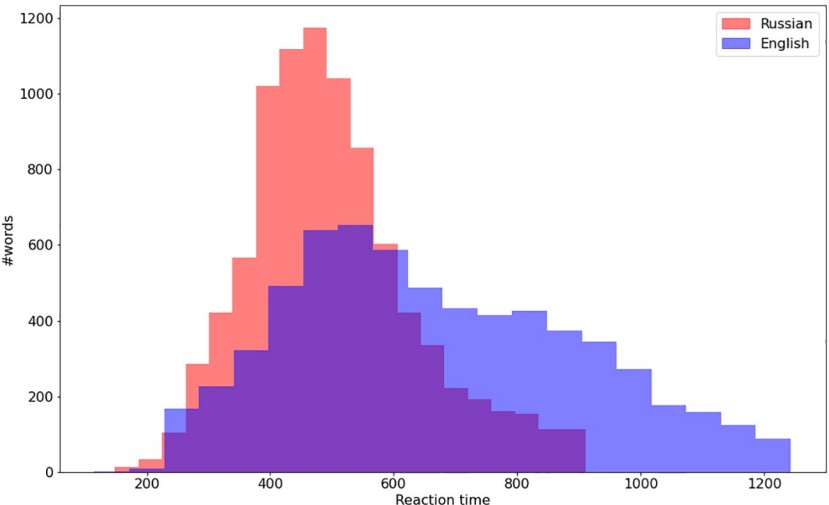

**Fig 3. Reaction time distribution for English and Russian words.**

prediction of word AUROC were close to zero, the AUROCs of participant-independent models for hold-out set were close to 0.5 with small standard deviation, suggesting that the model did not learn anything useful for prediction of experiment outcome.

## 4 Discussion

The aim of the current study was to replicate and extend the findings of [20](Experiment 1) with the VLPFC and DLPFC. However, the effect of tDCS over the left VLPFC (decrease in false recognition) was not replicated, and there were weak, negative effects over the DLPFC in the Offline group. In fact, there was a trend toward increased false alarms after online tDCS over the VLPFC and offline tDCS over the DLPFC compared to Sham. The trend suggests that an increase in false alarms obscured group differences, and the increase in false alarms could be driven by an increase in semantic elaboration that leads to better memory for features common to multiple items [45, 46]. While the results do not support effects over the DLPFC and VLPFC, this speaks to the lack of tDCS effect rather than the lack of involvement of these regions in episodic memory. Although it is clear from TMS studies that the VLPFC is necessary for episodic memory, TMS disruption of the DLPFC does not consistently impair episodic memory and the specific roles of both regions remain to be clarified. While meta-analyses and studies aimed at replication of tDCS cognitive effects do not support the effectiveness of tDCS generally [47, 48], the majority of tDCS studies lack deeper explorations of non-linear effects and individual differences through modelling and ML [49]. In our case, a limitation was not estimating whether maximum current flow was underneath the anode electrode and at the region of interest (VLPFC or DLPFC). However, the aim of the current study was to clarify the consistency of the effect found in the original study using an identical method, including montage. Future studies could test a montage with greatest likelihood of current flow at the region of interest by modeling current density maxima. More insight is needed into the relationship between biological and cognitive effects. For example, future studies could attempt to classify groups by using concurrent EEG activity during tDCS administration. However, even tDCS-EEG studies alone may not be beneficial because EEG reflects the synchronized firing of large populations of neurons

across the brain (not just the stimulated region), and the neural signature is not always modulated by changes in cognitive performance. Thus, more sensitive measures of tDCS cognitive and neural effects can be implemented, such as changes in resting-state connectivity [50], GABA and glutamate concentrations [51], and cerebral blood flow [52]. It remains unclear whether the significant tDCS modulation of long-term verbal encoding in the original study occurred through modulation of related processes such as lexical retrieval, working memory, or attention. For example, tDCS has been shown to modulate the speed of vocabulary learning but not final vocabulary recall over multiple sessions [53]. Moreover, atDCS over the VLPFC led to improved semantic processing in language tasks [32] such as naming common objects [54] and working memory tasks [55]. Thus, tDCS could modulate a semantic process or working memory maintenance that contributes to long-term memory rather than the formation of the memory trace, which is thought to rely more on the hippocampus [56, 57]. Individual differences in baseline memory performance, correlated cognitive functions, and neuroanatomy could obscure the effect of tDCS on measures of memory performance, especially if the effect is subtle. It is clear that effects of tDCS on physiology may not correspond with cognitive effects, and future studies can better explore this relationship with physiological as well as cognitive measures and by estimating the current density maxima. It is notable that using AUROC we could not discriminate hard and easy to remember words in Experiment 1 of [20]: the AUC curve revealed no significant effect of VLPFC tDCS on English or Russian speakers. Although average memory accuracy was above chance, AUC suggested that when considering individual trials, participants were guessing. Future studies can test the reliability of the recognition test as an assessment for tDCS-induced cognitive effects and try to increase participant motivation and concentration to attain optimal trial-by-trial performance for examining tDCS-induced changes. The ML analysis showed that phonological, semantic, and orthographic features did not influence the episodic memory. Although it is likely that these features contributed to memory formation, their influence was minimal and overshadowed by the large interaction between atDCS and individual differences. This individual variability has been examined in other studies that found differences between high and low performers (e.g. [10]). Individual differences in baseline memory performance and encoding ability appear to be an influential factor, more than language or word characteristics. The results indicate that individual differences may be one of the most important parameters not only in the current study and tDCS field but also in replication studies. A limitation was that we did not conduct a within-subjects study in which we examined differences in individual performance. A replication of a within-subjects study could reveal that the effect of a between-subjects study was due to differences between participants and when participants are compared to their own performance, there is little effect of tDCS. Indeed, studies including [20] have found varying effects when comparing between-subjects to within-subjects tDCS effects over the same location, with potentially smaller or absent effects for within-subjects studies (e.g. compare between-subjects results in [30, 32, 58, 59] and Experiment 1 in [20] to within-subjects results in Experiment 3 in [20]; Experiment 2 in [60] and [14]. Furthermore, it is important to identify the reliability of tDCS effects with direct replications within the same lab as well as other labs [61] with sufficiently large sample sizes. However, continued future examinations of tDCS cognitive effects may have less value if not supplemented with a measure of biological effects. Future data-driven studies should aim to predict when physiological effects such as increases in BOLD activation lead to cognitive outcomes such as higher performance and whether duration and frequency of tDCS are involved. Although the sample size of the current study should have had sufficient statistical power to detect the original effect size ([20], Experiment 1), the true effect size could still be eluded because of a statistical phenomenon known as the "winner's curse". The

winner's curse posits that the first studies to find a significant and novel effect will be published, and the reported effect will be exaggerated because these studies tend to be exploratory and include smaller sample sizes. If several small-sample studies are investigating the same effect, random error and sampling variation may lead to one but not all of the studies finding an effect that crosses significance threshold because under-powered studies can only find large effects. The true effect is likely to be smaller, so it would not otherwise emerge in under-powered studies. This phenomenon is illustrated well in the meta-analysis by [2], in which most included articles reported at least one significant effect, but the average effect size was close to zero when all studies were included in the analysis. For example, [62] found an effect of atDCS in their first experiment but not in a subsequent replication in the same paper. The conclusion of [63] is in line with the "winner's curse" effect, while the authors suggest a smaller sample size (between $d = .40$ and $d = .50$ according to Cohen, 1988) as more appropriate for tDCS studies. We did not successfully replicate the results of the previous work, although we used a larger sample size and nearly identical method. It is worth mentioning that the results of the replicated study are not significant based on the ANOVA (and this statistical model is appropriate for the original study) or the AUROC analysis applied to both datasets. It does not seem that there was a cultural or linguistic component involved, in line with previous tDCS experiments that found similar effects on verbal memory performance between countries (Italy: [32]; England: [20]; USA: [59]). However, culture-dependent tDCS should be tested directly by comparing individuals of different cultures in the same language. We would expect culture-dependent effects in social cognition (e.g. [53]) but not in processes such as memory encoding that are thought to rely on the same neural architecture across people. In the current study, tDCS did not lead to significant differences in memory performance or reaction time. While it seems unlikely that the significant effect in the original study was due to sample size alone, we conclude based on the lack of replication that the effect of tDCS on long-term verbal memory when the anode is placed over the VLPFC or DLPFC may be subtle. Moreover, the effect may be completely obscured during interactions with other factors that may be difficult to measure or quantify, such as the state of the network. Regression approaches could be successful for future studies that model interactions between biological and cognitive effects. In spite of the significant tDCS effects found in previous memory studies, we suggest caution in interpreting these effects and applying tDCS as a neuromodulator until replications are conducted with biomarkers.

## Supporting information

**S1 Appendix. Significantly different words.** Russian and English words that differ significantly in the number of hits, false alarms, AUROC and reaction time.
(PDF)

**S2 Appendix. Data augmentations.** The procedures used to augment the training set according to the method proused in [41].
(PDF)

**S3 Appendix. Participant-independent ML trials.** The performance of participant-independent TPOT and AutoPytorch models.
(PDF)

**S4 Appendix. Participant-dependent ML trials.** The performance of participant-independent TPOT and AutoPytorch models.
(PDF)

## Acknowledgments

The study used the HSE Synchronous Eye-tracking, Brain Signal Recording and Non-Invasive Brain Stimulation System. We are grateful for the assistance of Anastasiya Bogdanova and Daniil Krivenok in data collection.

## Author Contributions

**Conceptualization:** Aleksandra Petrovskaya, Giulia Galli, Matteo Feurra.

**Data curation:** Bogdan Kirillov, Giulia Galli, Angela Medvedeva.

**Formal analysis:** Bogdan Kirillov, Angela Medvedeva.

**Funding acquisition:** Matteo Feurra.

**Investigation:** Aleksandra Petrovskaya, Anastasiya Asmolova.

**Methodology:** Aleksandra Petrovskaya, Bogdan Kirillov, Giulia Galli.

**Project administration:** Aleksandra Petrovskaya, Anastasiya Asmolova, Angela Medvedeva.

**Resources:** Giulia Galli, Matteo Feurra.

**Software:** Bogdan Kirillov.

**Supervision:** Aleksandra Petrovskaya, Matteo Feurra, Angela Medvedeva.

**Validation:** Aleksandra Petrovskaya, Bogdan Kirillov, Angela Medvedeva.

**Visualization:** Bogdan Kirillov.

**Writing – original draft:** Aleksandra Petrovskaya, Bogdan Kirillov, Angela Medvedeva.

**Writing – review & editing:** Aleksandra Petrovskaya, Bogdan Kirillov, Giulia Galli, Angela Medvedeva.

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
