## [Decision Letter · Decision Letter 0]

20 Jul 2020

PONE-D-20-16976

Examining the effects of transcranial direct current stimulation on human episodic memory with machine learning

PLOS ONE

Dear Dr. Kirillov,

Thank you for submitting your manuscript to PLOS ONE. After careful consideration by 2 Reviewers and an Academic Editor, all of the critiques of both Reviewers must be addressed in detail in a revision to determine publication status. If you are prepared to undertake the work required, I would be pleased to reconsider my decision, but revision of the original submission without directly addressing the critiques of the two Reviewers does not guarantee acceptance for publication in PLOS ONE. If the authors do not feel that the queries can be addressed, please consider submitting to another publication medium. A revised submission will be sent out for re-review. The authors are urged to have the manuscript given a hard copyedit for syntax and grammar.

**Comments to the Author**

1. Is the manuscript technically sound, and do the data support the conclusions?

Reviewer #1: Yes

Reviewer #2: Yes

2. Has the statistical analysis been performed appropriately and rigorously? 

Reviewer #1: Yes

Reviewer #2: Yes

3. Have the authors made all data underlying the findings in their manuscript fully available?

Reviewer #1: Yes

Reviewer #2: Yes

4. Is the manuscript presented in an intelligible fashion and written in standard English?

Reviewer #1: Yes

Reviewer #2: Yes

5. Review Comments to the Author

Reviewer #1: The aim of the this study conducted by Petrovskaya et al. was to use tDCS to replicate and extend the findings of Medvedeva et al. (2019)](Experiment 1) with the anode over VLPFC and DLPFC. However, the effect of tDCS with the anode over the left VLPFC (decrease in false recognition) was not replicated, and there were weak, negative effects with the anode over the DLPFC in the Offline group. In fact, there was a trend toward increased false 309 alarms after online active tDCS over the VLPFC and offline active tDCS over the DLPFC compared to Sham. The trend suggests that an increase in false alarms obscured group differences, and the increase in false alarms could be driven by an increase in semantic elaboration that leads to better memory for features common to multiple items. While the results do not support effects of tDCS with the anode over the DLPFC and VLPFC, this speaks to the lack of tDCS effect rather than the lack of involvement of these regions in episodic memory.

The manuscript is well written and I like the approach to investigate cognitive effects with machine learning, but there are some issues that need to be addressed.

The authors stated that tDCS was applied over VLPFC and DLPFC but only the anode was placed over these brain areas. I do not see any evidence that this electrode montage targeted the VLPFC and DLPFC. The current density maxima seems to be between the electrodes. Please provide evidence that the montage targets VLPFC and DLPFC. For instance, HD-target software (Soterix medical) use to determine the optimal electrode configuration for your chosen brain target shows that in order to target the Left DLPFC the anode should be over C3 and the cathode over AF3 according to 10-20 EEG electrode positioning system. If you do not have evidence you should mention this issue as a limitation.

Please state that the anode was applied over VLPFC and DLPFC instead of anodal tDCS.

Reviewer #2: This is a well-written replication study of the effect of tDCS on memory encoding. Such studies are important to verify the claims in the brain stimulation literature, particularly as there have been some criticisms of the field in recent years (e.g. Horvath et al. meta-analysis cited by the authors of the present study). My only major concern is regarding the interpretation of results:

Page 10, Line 330 – “Nonetheless, the results of the current study support the conclusion that tDCS does not modulate memory in the VLPFC and DLPFC directly but rather a subprocess leading to successful memory formation and retrieval.”

It's not clear that this can be claimed on the basis of the present findings. There was no direct examination of subprocesses leading to successful memory formation and retrieval. Rather, results appear to largely conform with the null hypothesis. Indeed, the significant ANOVA findings for false alarms could be somewhat questioned, given multiple ANOVAs were conducted (7 in total), which can inflate the false positive rate in much the same way that conducting multiple pairwise comparisons can. Correcting for multiple tests, by using a MANOVA for example, would likely leave the false alarm results non-significant. If this is indeed the case, one would have to conclude the the current study failed to replicate the previous findings, that the stimulation conditions did not significantly differ on any of the outcome variables, and that tDCS to the DLPFC or VLPFC was not observed to improve memory encoding.

Page 11, Line 400 – “We conclude that tDCS may exert a subtle modulation that also interacts strongly with individual differences, particularly in baseline activation and neuroanatomy.”

Similar to previous point, it’s not clear what empirical data obtained from the current study leads to the conclusion that individual differences, including baseline activity and neuroanatomy, may alter findings for tDCS and memory encoding. While this can be presented speculatively as an important variable/s to consider and investigate in future research, it’s not clear that these factors are at all relevant to the current findings. It may well be possible, at least theoretically, that they are irrelevant and tDCS exerts little to no reliable effect on memory encoding in healthy individuals.

Minor issues

Abstract – sample size would be useful information to include in the Abstract.

Abstract – Acronyms for VLPFC and DLPFC not explicitly defined in the abstract.

Page 1, Line 6 – a citation is needed to support the claim that tDCS can decrease reaction time or increase memory accuracy.

Page 2, Line 35 – “…and the left hemisphere in the VLPFC appears selectively engaged…” Somewhat oddly worded, could be rephrased to the left VLPFC

Page 3, Line 42 – “In one tDCS study, [27] found that cathodal stimulation over the left DLPFC (F3) in a cued-word-completion learning task where errors were evoked through guessing the wrong completion lowered memory accuracy in contrast to a non-error condition or anodal stimulation in either condition (no effect).”

This sentence is difficult to parse, and may need to be presented in two sentences to better convey the meaning.

Page 3, Line 51 – “The current study involved administering atDCS over the left DLPFC offline or online encoding”. I believe there is a missing word, ‘during’, between DLPFC and offline.

Methods – A few additional details should be provided in the Methods. What was the specific timing of the stimulation relative to the task in the offline condition (e.g. did one immediately follow the other)? What was the duration of stimulation for active tDCS? This is described in citation [1], but could be repeated in the present manuscript.

Page 4, Line 113 – Minor typo of ‘ae’ instead of ‘age in “Bonferroni-corrected pairwise comparisons of ae…”

Considering the significant difference in age between two of the groups (VLPFC online and DLPFC online), the authors could run one-way ANCOVAs, incorporating Age as a covariate, to ensure that this variable doesn’t confound the results.

There is some repetition in the Methods that could be minimised:

Page 5, Line 96 – “However, stimulation was delivered only on the first day and at different sites (DLPFC or VLPFC) and times (offline or online encoding) depending on the group.”

Page 4, Line 137 – “Stimulation was only delivered on the first day, but electrodes were placed on the head on the second day without stimulation (to replicate the method of [1] exactly).”

Page 4, Line 139 – “For all groups, stimulation was delivered on the first day only.”

Page 6, Line 218 – A brief description of what is meant by ‘letter-based one-hot encoding’ would be useful.

6. PLOS authors have the option to publish the peer review history of their article (what does this mean?). If published, this will include your full peer review and any attached files.

**Do you want your identity to be public for this peer review?** For information about this choice, including consent withdrawal, please see our Privacy Policy.

Reviewer #1: No

Reviewer #2: **Yes: **Stevan Nikolin

We look forward to receiving your revised manuscript.

Kind regards,

Stephen D. Ginsberg, Ph.D.

Section Editor

PLOS ONE

2. Your abstract cannot contain citations. Please only include citations in the body text of the manuscript, and ensure that they remain in ascending numerical order on first mention.

3. Please ensure that you refer to Figure 3 in your text as, if accepted, production will need this reference to link the reader to the figure.

4. We note you have included a table to which you do not refer in the text of your manuscript. Please ensure that you refer to Table 1 in your text; if accepted, production will need this reference to link the reader to the Table.

---

## [Author Response · Author response to Decision Letter 0]

18 Nov 2020

Responses to reviewer #1:

Reviewer #1: The aim of the this study conducted by Petrovskaya et al. was to use tDCS to replicate and extend the findings of Medvedeva et al. (2019)](Experiment 1) with the anode over VLPFC and DLPFC. However, the effect of tDCS with the anode over the left VLPFC (decrease in false recognition) was not replicated, and there were weak, negative effects with the anode over the DLPFC in the Offline group. In fact, there was a trend toward increased false 309 alarms after online active tDCS over the VLPFC and offline active tDCS over the DLPFC compared to Sham. The trend suggests that an increase in false alarms obscured group differences, and the increase in false alarms could be driven by an increase in semantic elaboration that leads to better memory for features common to multiple items. While the results do not support effects of tDCS with the anode over the DLPFC and VLPFC, this speaks to the lack of tDCS effect rather than the lack of involvement of these regions in episodic memory.

The manuscript is well written and I like the approach to investigate cognitive effects with machine learning, but there are some issues that need to be addressed.

The authors stated that tDCS was applied over VLPFC and DLPFC but only the anode was placed over these brain areas. I do not see any evidence that this electrode montage targeted the VLPFC and DLPFC. The current density maxima seems to be between the electrodes. Please provide evidence that the montage targets VLPFC and DLPFC. For instance, HD-target software (Soterix medical) use to determine the optimal electrode configuration for your chosen brain target shows that in order to target the Left DLPFC the anode should be over C3 and the cathode over AF3 according to 10-20 EEG electrode positioning system. If you do not have evidence you should mention this issue as a limitation.

Please state that the anode was applied over VLPFC and DLPFC instead of anodal tDCS.

Response: We are glad the reviewer finds the approach and manuscript suitable and thank the reviewer for pointing this out as an important consideration. Modelling approaches are valuable for confirming that the stimulation corresponds to the expected montage. We have stated a lack of evidence for our montage targeting the VLPFC and DLPFC as a limitation. We have also corrected the description of the montage to state “the anode was applied” rather than “anodal tDCS.” 

Throughout the manuscript, atDCS was replaced with tDCS in reference to the current study.

Page 4 lines 95-97

Specifically, we applied the same tDCS settings (2 mA anodal active tDCS and 30 s Sham tDCS) over the DLPFC (all conditions: offline, sham, and online) in addition to the VLPFC (online encoding only). 

Page 10 lines 342-348

In our case, a limitation was not estimating whether maximum current flow was underneath the anode electrode and at the region of interest (VLPFC or DLPFC). However, the aim of the current study was to clarify the consistency of the effect found in the original study using an identical method, including montage. Future studies could test a montage with greatest likelihood of current flow at the region of interest by modeling current density maxima. 

Reviewer #2: This is a well-written replication study of the effect of tDCS on memory encoding. Such studies are important to verify the claims in the brain stimulation literature, particularly as there have been some criticisms of the field in recent years (e.g. Horvath et al. meta-analysis cited by the authors of the present study). My only major concern is regarding the interpretation of results:

Page 10, Line 330 – “Nonetheless, the results of the current study support the conclusion that tDCS does not modulate memory in the VLPFC and DLPFC directly but rather a subprocess leading to successful memory formation and retrieval.”

It's not clear that this can be claimed on the basis of the present findings. There was no direct examination of subprocesses leading to successful memory formation and retrieval. Rather, results appear to largely conform with the null hypothesis. Indeed, the significant ANOVA findings for false alarms could be somewhat questioned, given multiple ANOVAs were conducted (7 in total), which can inflate the false positive rate in much the same way that conducting multiple pairwise comparisons can. Correcting for multiple tests, by using a MANOVA for example, would likely leave the false alarm results non-significant. If this is indeed the case, one would have to conclude the the current study failed to replicate the previous findings, that the stimulation conditions did not significantly differ on any of the outcome variables, and that tDCS to the DLPFC or VLPFC was not observed to improve memory encoding.

Page 11, Line 400 – “We conclude that tDCS may exert a subtle modulation that also interacts strongly with individual differences, particularly in baseline activation and neuroanatomy.”

Similar to previous point, it’s not clear what empirical data obtained from the current study leads to the conclusion that individual differences, including baseline activity and neuroanatomy, may alter findings for tDCS and memory encoding. While this can be presented speculatively as an important variable/s to consider and investigate in future research, it’s not clear that these factors are at all relevant to the current findings. It may well be possible, at least theoretically, that they are irrelevant and tDCS exerts little to no reliable effect on memory encoding in healthy individuals.

Response: We thank the reviewer for the thoughtful consideration of the manuscript and helpful suggestions for clarifying and resolving certain issues. We added a correction for multiple ANOVAs using a MANOVA and tested group differences in recognition accuracy and reaction time with age as a covariate. In line with these results that show no differences between groups, we clarify in the discussion that there may be no significant effects of tDCS on memory performance in the current study. We clarify that while individual differences may interact with tDCS to influence cognitive effects, further work is needed to explore this relationship. 

Page 7 Lines 270-277

To enable comparison with the original paper, we report the results of multiple ANOVAs. However, to control for multiple tests and the difference in age between groups, we analyzed all dependent variables including AUROC with a MANOVA, Pillai’s trace (12, 294) = .166, p = .148, and recognition accuracy and average reaction time with age as a covariate, F(3, 98) = .097 and F(3, 98) = .557, respectively. There was no significant difference between groups with hits, false alarms, discrimination ability, and average reaction time as dependent variables. 

Page 11

Lines 362-364

It remains unclear whether the significant tDCS modulation of long-term verbal encoding in the original study occurred through modulation of related processes such as lexical retrieval, working memory, or attention. 

Lines 370-376

Individual differences in baseline memory performance, correlated cognitive functions, and neuroanatomy could obscure the effect of tDCS on measures of performance, especially if the effect is subtle. It is clear that effects of tDCS on physiology may not correspond with cognitive effects, and future studies can better explore this relationship with physiological as well as cognitive measures and by estimating the current density maxima. 

Page 12

Lines 437-443

In the current study, tDCS did not lead to significant differences in memory performance or reaction time. While it seems unlikely that the significant effect in the original study was due to sample size alone, we conclude based on the lack of replication that the effect of tDCS on long-term verbal memory when the anode is placed over the VLPFC or DLPFC may be subtle. Moreover, the effect may be completely obscured during interactions with other factors that may be difficult to measure or quantify, such as the state of the network. 

Minor issues

We present parts of the manuscript that address each comment.

Abstract – sample size would be useful information to include in the Abstract.

Abstract – Acronyms for VLPFC and DLPFC not explicitly defined in the abstract.

We aimed to replicate a published effect of transcranial direct-current stimulation (tDCS)-induced recognition enhancement over the human ventrolateral prefrontal cortex (VLPFC) [1] and analyse the data with machine learning. We investigated effects over an adjacent region, the dorsolateral prefrontal cortex (DLPFC). In total, we analyzed data from 97 participants after exclusions. We found weak or absent effects over the VLPFC and DLPFC. We conducted machine learning studies to examine the effects of semantic and phonetic features on memorization, which revealed no effect of VLPFC tDCS on the original dataset or the current data. The highest contributing factor to memory performance was individual differences in memory not explained by word features, tDCS group, or sample size, while semantic, phonetic, and orthographic word characteristics did not contribute significantly. To our knowledge, this is the first tDCS study to investigate cognitive effects with machine learning, and future studies may benefit from studying physiological as well as cognitive effects with data-driven approaches and computational models.

Page 2, Line 6 – a citation is needed to support the claim that tDCS can decrease reaction time or increase memory accuracy.

Transcranial direct-current stimulation (tDCS), a method of safely and non-invasively delivering a weak electric current through the cortex, has been gaining increasing attention [2] as a tool for studying and possibly enhancing episodic memory [3]. Anodal tDCS (atDCS) involves increasing cortical excitability in a target region such as the dorsolateral prefrontal cortex (DLPFC), which may in turn facilitate or enhance memory performance by decreasing reaction time and/or increasing memory accuracy. (3,4)

3. Investigating the Role of Current Strength in tDCS Modulation of Working Memory Performance in Healthy Controls 

Florence Teo,1 Kate E. Hoy,1,* Zafiris J. Daskalakis,2 and Paul B. Fitzgerald1

Front Psychiatry. 2011; 2: 45. Published online 2011 Jul 18. doi: 10.3389/fpsyt.2011.00045

PMCID: PMC3141358

PMID: 21811474

4. Shaping Memory Accuracy by Left Prefrontal Transcranial Direct Current Stimulation

Bastian Zwissler,1 Christoph Sperber,1 Sina Aigeldinger,1 Sebastian Schindler,2 Johanna Kissler,2 and Christian Plewnia,3

J Neurosci. 2014 Mar 12; 34(11): 4022–4026.

doi: 10.1523/JNEUROSCI.5407-13.2014

Page 2, Line 35 – “…and the left hemisphere in the VLPFC appears selectively engaged…” Somewhat oddly worded, could be rephrased to the left VLPFC

The VLPFC and DLPFC are thought to play functionally distinct roles in long-term verbal memory, and the left VLPFC appears selectively engaged in verbal but not non-verbal material.

Page 3, Line 42 – “In one tDCS study, [27] found that cathodal stimulation over the left DLPFC (F3) in a cued-word-completion learning task where errors were evoked through guessing the wrong completion lowered memory accuracy in contrast to a non-error condition or anodal stimulation in either condition (no effect).”

This sentence is difficult to parse, and may need to be presented in two sentences to better convey the meaning.

In a tDCS study conducted by [27], tDCS was applied over the left DLPFC in a cued-word-completion task to modulate memory performance. The results showed that cathodal stimulation inhibited memory accuracy after error learning in comparison with sham. The errors were evoked through guessing, or incorrect completion lowered memory accuracy in contrast to a non-error condition or anodal stimulation in either condition (no effect).

Page 3, Line 51 – “The current study involved administering atDCS over the left DLPFC offline or online encoding”. I believe there is a missing word, ‘during’, between DLPFC and offline.

The current study involved administering atDCS over the left DLPFC during offline or online encoding

Methods – A few additional details should be provided in the Methods. What was the specific timing of the stimulation relative to the task in the offline condition (e.g. did one immediately follow the other)? What was the duration of stimulation for active tDCS? This is described in citation [1], but could be repeated in the present manuscript.

Page 5

Lines 149-154

Stimulation was applied for the entire duration (10 minutes) of the reading task for the DLPFC offline group and for the entire duration (10 minutes) of the study memory task that immediately followed for the VLPFC Online and DLPFC Online groups. In summary, the stimulation was delivered for approximately 10 minutes to cover the entire duration of the reading and memory tasks. 

Page 4, Line 113 – Minor typo of ‘ae’ instead of ‘age in “Bonferroni-corrected pairwise comparisons of ae…”

Bonferroni-corrected pairwise comparisons of age

Considering the significant difference in age between two of the groups (VLPFC online and DLPFC online), the authors could run one-way ANCOVAs, incorporating Age as a covariate, to ensure that this variable doesn’t confound the results. 

As cited in an earlier response, we conducted an ANCOVA in addition to a MANOVA:

Page 7 Lines 270-277

To enable comparison with the original paper, we report the results of multiple ANOVAs. However, to control for multiple tests and the difference in age between groups, we analyzed all dependent variables including AUROC with a MANOVA, Pillai’s trace (12, 294) = .166, p = .148, and recognition accuracy and average reaction time with age as a covariate (ANCOVA), F(3, 98) = .097 and F(3, 98) = .557, respectively. There was no significant difference between groups with hits, false alarms, discrimination ability, and average reaction time as dependent variables.

There is some repetition in the Methods that could be minimised:

Page 5, Line 96 – “However, stimulation was delivered only on the first day and at different sites (DLPFC or VLPFC) and times (offline or online encoding) depending on the group.”

Page 4, Line 137 – “Stimulation was only delivered on the first day, but electrodes were placed on the head on the second day without stimulation (to replicate the method of [1] exactly).”

Page 4, Line 139 – “For all groups, stimulation was delivered on the first day only.”

We minimized repetition and increased conciseness in the method:

Page 4 Lines 97-101

The electrodes were placed at both sessions (study and test 24 hours later), but stimulation was only delivered before or during the encoding task in the study phase. The anode was placed over the left VLPFC (F7) or the left DLPFC (F3), and the cathode was placed over the ipsilateral (left) shoulder. See section 2.2 below for the experimental conditions.

Page 6, Line 218 – A brief description of what is meant by ‘letter-based one-hot encoding’ would be useful.

Lines 227-229

In line with the trial-based and participant-based approaches, we conducted two kinds of ML studies: 1) Participant-independent (trial-based) analysis tests the success of a model that predicts the AUROC for each word using either word embeddings or letter-based one-hot encoding (each letter i is replaced by a vector of Nalphabet components with the i-th component being equal to 1 while every other component is zero) and 2) Participant-specific (participant-based) analysis tests the success of a model that predicts whether a participant was able to recall the word given either word embeddings or letter-based one-hot encoding for each word for individual participants.

---

## [Decision Letter · Decision Letter 1]

23 Nov 2020

Examining the effects of transcranial direct current stimulation on human episodic memory with machine learning

PONE-D-20-16976R1

Dear Dr. Kirillov,

We’re pleased to inform you that your manuscript has been judged scientifically suitable for publication and will be formally accepted for publication once it meets all outstanding technical requirements.

Kind regards,

Stephen D. Ginsberg, Ph.D.

Section Editor

PLOS ONE

**Comments to the Author**

1. If the authors have adequately addressed your comments raised in a previous round of review and you feel that this manuscript is now acceptable for publication, you may indicate that here to bypass the “Comments to the Author” section, enter your conflict of interest statement in the “Confidential to Editor” section, and submit your "Accept" recommendation.

Reviewer #1: All comments have been addressed

2. Is the manuscript technically sound, and do the data support the conclusions?

Reviewer #1: Yes

3. Has the statistical analysis been performed appropriately and rigorously? 

Reviewer #1: Yes

4. Have the authors made all data underlying the findings in their manuscript fully available?

Reviewer #1: Yes

5. Is the manuscript presented in an intelligible fashion and written in standard English?

Reviewer #1: Yes

6. Review Comments to the Author

Reviewer #1: The authors addressed all my concerns. Their response yo my questions was hood. I think the paper is ready for publication

7. PLOS authors have the option to publish the peer review history of their article (what does this mean?). If published, this will include your full peer review and any attached files.

Reviewer #1: No

---

## [Editor Report · Acceptance letter]

26 Nov 2020

PONE-D-20-16976R1 

Examining the effects of transcranial direct current stimulation on human episodic memory with machine learning 

Dear Dr. Kirillov:

I'm pleased to inform you that your manuscript has been deemed suitable for publication in PLOS ONE. Congratulations! Your manuscript is now with our production department. 

Kind regards, 

on behalf of

Dr. Stephen D. Ginsberg 

Section Editor

PLOS ONE